# Biological Nanocarriers in Cancer Therapy: Cutting Edge Innovations in Precision Drug Delivery

**DOI:** 10.3390/biom15060802

**Published:** 2025-05-31

**Authors:** Ramesh Ganpisetti, Sanjay Giridharan, G. S. Sainaga Jyothi Vaskuri, Nikesh Narang, Pratap Basim, Mehmet Remzi Dokmeci, Menekse Ermis, Satish Rojekar, Amol D. Gholap, Nagavendra Kommineni

**Affiliations:** 1Terasaki Institute of Biomedical Sciences, 21100 Erwin St, Woodland Hills, Los Angeles, CA 91367-3712, USA; ramesh.ganpisetti@terasaki.org (R.G.); mdokmeci@terasaki.org (M.R.D.); mermis@terasaki.org (M.E.); 2Arizona State University, 1151 S. Forest AV Tempe, Tempe, AZ 85281, USA; sgiridh6@asu.edu; 3Department of Pharmaceutical Sciences, University of Tennessee Health Science Center (UTHSC), Memphis, TN 38163, USA; satyajyothi949@gmail.com; 4Department of Ophthalmology and Visual Sciences, University of New Mexico School of Medicine, Albuquerque, NM 87131, USA; nikesh_narang@yahoo.com; 5Thermo Fisher Scientific Inc., Cincinnati, OH 45237, USA; pratapbasim@gmail.com; 6Department of Pharmacological Sciences, Icahn School of Medicine at Mount Sinai, New York, NY 10029, USA; rojekarsatish@gmail.com; 7Department of Pharmaceutics, St. John Institute of Pharmacy and Research, Palghar 401404, Maharashtra, India; amolgholap16@gmail.com

**Keywords:** nanomedicine, cancer, precision, novel, chemotherapy, gene delivery

## Abstract

Cancer is a highly detrimental and fatal illness that poses a significant threat to human well-being. The pattern of cancer treatment is continuously being optimized by the advancement of old treatment approaches and the invention of novel treatment modes. Nanotechnology-based approaches are emerging as powerful candidates in the development of these advanced methods for treating tumors. This article provides a concise overview of nanotechnology for cancer or cancer nanomedicine and its applications. In light of prevalent issues, such as inadequate precision in targeting initial chemotherapy drugs, susceptibility of nucleic acid drugs to degradation, gene delivery, and the occurrence of common immune-related adverse events during immunotherapy, we explore the potential integration of nanomedicine with these treatment approaches and illustrative examples and highlight the benefits that arise from the utilization of nanomedicine.

## 1. Introduction

According to statistics compiled by the World Health Organization, cancer is identified as the leading cause of worldwide mortality, accounting for around 10 million deaths in 2022 and 20 million new cases of cancer [1]. Amongst different types of cancers diagnosed, the highest percentage was for lung, followed by breast, colorectal, prostate, stomach, and other types of cancers (represented in Figure 1) [1]. As per the latest data from National Cancer Institute (last assessed 28 April 2025), the new cancer cases in United States in 2025 include breast: 319,750 (16%), prostate: 313,780 (15%), lung and bronchus: 226,650 (11%), colon and rectum: 154,270 (8%), and other: 1,027,460 (50%) [2].

Projections indicate that the annual death toll from cancer will reach 70 million by the year 2050, a 77% increase from the incidence in 2022 [1]. Timely detection has a crucial role in combating cancer. The unfavorable outlook of cancer is mainly ascribed to the paucity of early diagnostic tools and the ineffectiveness of therapy [3]. Presently, the diagnosis and treatment approaches for cancer comprise using standardized screening techniques for a limited range of cancer types, followed by treatment that includes surgical procedures, radiation therapy, and chemotherapy [4]. Furthermore, alongside these conventional treatments, there has been extensive research conducted on immunotherapy, photothermal therapy, hormone therapy, gene therapy, and stem cell therapy in recent years.

Although there have been significant advancements in biology and clinical practices, chemotherapy remains the most efficient and cost-effective alternative among the various options. Chemotherapeutic drugs can cause dose-dependent side effects that may hinder treatment continuation and adversely affect a patient’s quality of life [5]. Furthermore, the majority of chemotherapeutic drugs exhibit inadequate solubility in water, leading to difficulties in formulating them, low bioavailability, and negative characteristics in terms of their distribution and clearance from the body. Another notable concern is the tendency of cancer cells to frequently acquire resistance to chemotherapy [6]. The primary concern for targeted drug delivery is the large size of the drug, which impacts its ability to cross the biological barriers. Potential solutions to these primary issues may involve the development of novel drug delivery systems that can precisely target medications to specific anatomical regions. Nanotechnology has demonstrated the ability to overcome several constraints of conventional chemotherapy to a certain degree, like mistargeting to healthy cells along with cancer cells, making it a valuable tool for enhancing the overall effectiveness of cancer treatment. Therapies based on nanoparticles possess a wide range of physical, chemical, and biological characteristics, such as a high surface area to volume ratio and the ability to navigate through cellular or tissue barriers, including the blood-brain barrier. These nanoparticles (NPs) can also carry specific agents on their surface, such as targeting ligands (e.g., antibodies, aptamers), for active targeting or surface modifications like PEGylation to prolong circulation and evade the reticuloendothelial system (RES) clearance. This feature allows nanoparticles to be functionalized with multiple surface biomolecules, such as immunoglobulins, small molecule drugs, aptamers, and peptides, thereby enabling multi-modal targeting and enhancing their therapeutic efficacy [7]. Nanotechnology-assisted molecular diagnostics greatly simplify cancer biomarker diagnostics in comparison to previous approaches. A nano biosensor enables the rapid identification of several protein biomarkers within a short timeframe, often within seconds. Nanotechnology-based cancer therapies offer benefits such as improved drug delivery, enhanced treatment efficacy, and reduced adverse effects. These nanoparticles can serve as carriers that can effectively bypass various biological barriers [7].

As elaborated earlier, various preclinical models have validated that nanocarriers enhance tumor-specific drug accumulation and prolong circulation, while also minimizing systemic degradation and toxicity [8,9,10]. Furthermore, they can inhibit the growth of solid tumors by enhancing other therapeutic strategies, such as photodynamic therapy (PDT), radiotherapies (RT), photothermal therapy (PTT), etc. [11]. Lastly, nanocarriers can reduce the toxicities associated with chemotherapeutic drugs by controlled yet sustained release and maintaining the drug release in therapeutic range of the drug with minimal toxicity. Recent advancements in nanotechnology have revealed promising potential in the field of cancer therapeutics. These advancements include the following emerging characteristics: (1) the ability to respond to stimuli [12], transform, or mimic biological properties through cell membrane coating [13]; (2) the combined administration of prodrugs, co-assembly drugs, or cell-nanocarrier conjugates; (3) the use of nanorobots or nano-catalysis reactions within the body [14]; and/or (4) enhanced immunotherapies in conjunction with nano vaccines to combat malignancies [15].

Anti-tumor nanotherapeutics have been rapidly advancing in basic research, leading to the invention of several nanodrugs that are now being used in clinical applications. These include micelles, drug-bound albumin nanocarriers, nanocrystals of drugs, inorganic NPs, polymeric NPs, and traditional liposomes. Although cancer treatments show promise in revolutionizing the field, the number of nanotherapeutics that have been clinically approved is very small when compared to the number of current preclinical studies. Liposomes and polymeric nanoparticles (NPs) are the primary types of nanodrugs that have been approved for clinical cancer therapy. They are administered intravenously. Liposomes are lipid vesicles made up of a double layer of amphiphilic phospholipids, which possess exceptional biocompatibility and biodegradability. Polymer-based nanoparticles (NPs) are additional nano systems that can be used to incorporate pharmaceuticals either within their structure or by attaching them to their surface. Apart from liposomes and polymeric nanoparticles, polymeric micelles, nanogels, gold nanospheres, gold nanorods, and carbon nanotubes have also been extensively studied, each with a distinct delivery mechanism (Figure 2). As far as we know, all the nanodrugs that have been approved by the Food and Drug Administration (FDA) for cancer treatment rely on passive targeting in order to reach the tumor locations [16,17].

The limited translation of these cancer nanomedicines from bench to bedside can be attributed to numerous factors. Genetic variability contributes to the intricate nature of malignancies, resulting in potentially diverse reactions to nanotherapeutics. The intricate interplay between nanodrugs and biological settings might significantly restrict the therapeutic results due to the protein corona-coating on nanomaterials during blood circulation, subsequent immune clearance, and the nanotoxicity induced by poor targeting effectiveness on normal tissues. The physical qualities of nanoparticles (NPs), such as size, shape, composition, surface features, and targeting ligands, can have a considerable impact on their biological outcomes [18]. Also, one of the less often asked questions in nanoparticle design is the role of nanoparticles’ geometry in determining phagocytotic (by macrophages) and first-pass metabolism fate of the drug delivery system, which can greatly influence the sustained release of the drug. Further, the limited stability during storage, particularly for nucleic acid-based nanodrugs, as well as the complex production procedures of nanoparticles, may also impede their clinical usage [19,20,21]. This review provides an overview of cancer nanomedicine, focusing on its advanced translational applications in the current scenario.

## 2. Current Status of Cancer Nanomedicine

Nanoparticles have been widely explored for the delivery of cancer therapeutics. Among the various classes of nanoparticles, lipid and polymer-based nanoparticles have received the attention of the scientific fraternity, which includes solid lipid nanoparticles, liposomes, and polymeric nanoparticles. In addition to these, dendrimers, polymeric micelles, gold nanoparticles, and carbon dots have also been extensively investigated [22,23].

Liposomes are closed, bilayered structures with an aqueous cavity and one or more bilayer phospholipid membranes that are produced when phospholipids self-assemble in an aqueous medium. In the United States (1995) and the European Union (1996), Doxil^®^/Caelyx^®^ was the first stealth liposomal formulation to be licensed for cancer therapy [24,25]. When compared to free doxorubicin, Doxil^®^ reduced cardiotoxicity and myelotoxicity while employing a liposomal composition of HSPC:CL:MPEG 2000-DSPE (*w*/*w* 3:1:1, calculated molar ratio 3:2:0.9), achieving higher drug concentrations in tumors [26,27]. Solid lipid nanoparticles (SLNs) are a type of colloidal delivery system composed of solid lipids that range in size from 50 to 1000 nm at ambient temperature. SLNs are thought to be the perfect vehicle for cancer treatments because they allow for the precise and controlled delivery of drugs that are entrapped in the solid lipid matrix [28]. Polymeric nanoparticles are colloidal macromolecules made up of several monomers with a particular structural architecture. To achieve controlled drug release in the target, the drug is either encapsulated or bonded to the surface of the nanoparticle, forming a nanosphere or a nanocapsule [18,29]. At first, non-biodegradable polymers, including polyacrylamide, polymethylmethacrylate (PMMA), and polystyrene, were used to produce polymeric nanoparticles [30]. However, because they were difficult to remove from the system, their accumulation resulted in toxicity. Nowadays, biodegradable polymers that are known to improve drug release and biocompatibility while lowering toxicity include polylactic acid, poly(amino acids), chitosan, alginate, and albumin [31]. With the benefits of both polymeric and lipid-based nanoparticles, the combination of lipid and polymer-based nanoparticles is being studied. It combines the benefits of both carrier systems, including a high drug-loading capacity, stability, enhanced biocompatibility, rate-limiting controlled release, prolonged drug half-lives, and therapeutic efficacy, while reducing their disadvantages (lipid polymer hybrid nanoparticles: a custom-tailored next-generation approach for cancer therapeutics). Carbon dots (CDs) also gained much attention since they exhibit special optical characteristics and inherent theranostic qualities; they have been identified as interesting candidates in nanotheranostics for concurrent bioimaging and cancer treatment. They can be applied to targeted chemotherapy, photodynamic therapy, photothermal therapy, and bioimaging. Additionally, CDs can be used in conjunction with other treatments and conjugated with anticancer drugs to provide more potent chemotherapy [32].

Nanoparticles take advantage of the leaky nature of the blood vessels in tumor tissues. Nanoparticles can passively pass through the capillary endothelial barrier and enter the interstitial space because tumor blood arteries contain more fenestrations [33]. Vascular endothelial cells in healthy, non-tumorous tissues are closely packed together with tiny paracellular spaces between 5 and 10 nm. On the other hand, depending on the type of cancer, the distances between endothelial cells in tumor blood arteries might vary from 100 to 700 nm [34]. Additionally, solid tumors lack a functioning lymphatic system due to their disordered vascular architecture. The enhanced permeability and retention (EPR) effect, which permits the passive distribution and accumulation of liposomes into the tumor site, is caused by the combination of the leaky tumor vasculature and the restricted lymphatic drainage [35]. Decreased efficacy and/or off-target toxicity are some of the drawbacks of this passive targeting of nanoparticles, which solely depends on the pathophysiological characteristics at the tumor location. The alternative targeting strategy is active targeting, which interacts with tumor-specific markers to directly target tumor cells using molecular techniques. Targeting moieties, such as monoclonal antibodies, antibody fragments, or peptides, are typically conjugated to the surface of actively targeted nanocarriers [36]. This method shows promise as a cancer treatment technique. Active targeting makes use of particular pathological alterations in the tumor microenvironment, like numerous protein overexpressions. To optimize drug delivery, cells that overexpress these proteins can selectively uptake nanocarriers that target these markers [37].

Existing cancer treatments, including surgery, radiation, and chemotherapy, have limitations due to their inability to completely remove cancerous areas and the potential harm they may cause to healthy tissues. Nanotechnology offers the means to accurately and directly deliver chemotherapies to cancer cells and enhance the effectiveness of radiation-based and other current therapy methods. These factors may lead to a decreased risk for the patient and an increased likelihood of survival. The initial cancer medicine utilizing nanotechnology that received approval from the U.S. FDA is the PEGylated doxorubicin liposomal formulation, which was introduced to the market in 1995. Since then, there has been substantial progress in the development of nanotherapeutic formulations, resulting in the approval of many nanomedicines by the FDA and other regulatory authorities for the treatment of cancer (Figure 3). Currently, there are various types of nanomedicines being used (both in clinical and experimental research), including polymeric micelles, liposomes, drug-bound albumin nanocarriers, inorganic NPs, lipid NPs, and nanocrystals of drugs (as shown in Table 1). However, liposomes (both PEGylated and non-PEGylated) and other lipid-based nanoparticles still make up a substantial portion of the nanotherapeutics available in the market.

Most licensed medications rely on passive targeting, while a small number of nanocarriers capable of ligand-mediated active targeting are also being explored in clinical studies [38]. These specifically targeted nanoparticles were designed to enhance the way the agents are distributed and processed in the body. Regrettably, the clinical results revealed that the ligand-mediated targeting strategies to enhance effectiveness (compared to NPs without ligands) provided only a small improvement. This could be attributed to the adsorption of protein corona, which causes the ligands on NPs to be concealed [39]

Liposomes remain the dominant category of clinically licensed nanomedicines, mostly due to their excellent safety [39]. In addition to approved nanomedicines, the FDA has authorized several Investigational New Drug applications for nanoformulations in recent years (Table 2). The majority of these trials involve the use of conventional chemotherapeutics with a pre-existing nanoplatform.

**Table 1 biomolecules-15-00802-t001:** Approved cancer drug therapies based on nanotechnology.

Product	Nanoparticle Material	Company	Drug/ Mechanism	Approval (Year)	Indication	References
Hensify (NBTXR3)	Hafnium oxide nanoparticle	Nanobiotix (Paris, France)	Radiotherapy	EMA (2019)	Locally advanced soft tissue sarcoma (STS)	[40]
Pazenir	Nanoparticle-bound albumin	Ratiopharm GmbH (Ulm, Germany)	Paclitaxel	EMA (2019)	Metastatic breast cancer, metastatic adenocarcinoma of the pancreas, non-small cell lung cancer	[41,42,43,44]
Vyxeos	Liposome	Celator/Jazz Pharma (NJ, USA)	Cytarabine/Daunorubicin	FDA (2017) EMA (2018)	Acute myeloid leukemia	
Onivyde	Liposome	Merrimack Pharma (MA, USA)	Irinotecan	FDA (2015)	Pancreatic cancer, colorectal cancer	[42,43,44]
NanoTherm	Iron oxide nanoparticles	MagForce Nanotechnologies AG (Berlin, Germany)	Thermal ablation with magnetic field	EMA (2010, 2013)	Glioblastoma, prostate, and pancreatic cancer	
Marqibo	Liposome	Talon Therapeutics/Spectrum Pharmaceuticals (MA, USA)	Vincristine	FDA (2012)	Acute lymphoblastic leukemia	
Mepact	Liposome	Takeda Pharmaceuticals (Tokyo, Japan)	Mifamurtide MTP-PE	EMA (2009)	Osteosarcoma	[43,44]
Genexol-PM	PEG-PLA polymeric micelle	Samyang Biopharmaceuticals (Gyeonggi-do, South Korea)	Paclitaxel	South Korea (2007)	Breast, lung, ovarian cancer	
Oncaspar	Polymer protein conjugate	Les Laboratoires Servier (Suresnes, France)	Pegaspargase/L-asparaginase	FDA (1994, 2006)	Acute lymphoblastic leukemia	
Abraxane	Nanoparticle-bound albumin	Abraxis/Celgene (NJ, USA)	Paclitaxel	FDA (2005)	Breast and pancreatic cancer, non-small-cell lung cancer	
DepoCyt	Liposome	Pacira Pharmaceuticals (NJ, USA)	Cytarabine	FDA (1999)	Neoplastic meningitis	
DaunoXome	Liposome	Gilead Sciences (CA, USA)	Daunorubicin	FDA (1996)	Kaposi’s sarcoma	
Doxil	Liposome	Johnson and Johnson (NJ, USA)	Doxorubicin	FDA (1995, 1999, 2007), EMA (1996, 2000), Taiwan (1998)	Metastatic breast cancer, ovarian cancer, Kaposi’s sarcoma, multiple myeloma	
Lipusu	Liposome	Luye Pharma (China)	Paclitaxel	State Food and Drug Administration of China (2006)	NSCLC, ovarian cancer, and breast cancer	
DHP107	Lipid nanoparticle	Daehwa Pharmaceutical (Gangwon-do, South Korea)	Paclitaxel	South Korea (2016	Gastric cancer	
Apealea	Micelle	Oasmia Pharmaceutical (Uppsala, Sweden)	Paclitaxel	EMA (2018)	Ovarian, peritoneal, and fallopian tube cancer	

**Table 2 biomolecules-15-00802-t002:** Nanotechnology-based products under clinical trials.

Product	Nanoparticle Material	Company	Active Ingredient	Clinical Trial Number	Indication	References
Docetaxel-PNP	Polymeric nanoparticles	Samyang Biopharmaceuticals Corporation (Gyeonggi-do, South Korea)	Docetaxel	NCT01103791	Advanced solid malignancies	[41,45,46]
ABT-888	PEGylated liposomes	AbbVie (IL, USA)	Temozolomide and lipo somal and doxorubicin	NCT01113957	Ovarian cancer	[45,46]
LipoVNB	Liposomes	Taiwan Liposome Company (Taipei, Taiwan)	Vinorelbine tartrate	NCT02925000	Advanced malignancy	
FF-10850	Liposomes	Fujifilm Pharmaceuticals U.S.A., Inc. (MA, USA)	Topotecan	NCT04047251	Advanced solid tumors	
LY01610	Liposomes	Luye Pharma Group Ltd. China)	Irinotecan hydrochloride	NCT04381910	Small cell lung cancer	
Mitoxantrone hydrochloride liposome injection	Liposomes	CSPC ZhongQi Pharmaceutical Technology Co., Ltd. (Hebei, China)	Mitoxantrone hydrochloride	NCT04927481	Breast cancer	
WGI-0301	Lipid nanoparticles	Zhejiang Haichang Biotech Co., Ltd. (Zhejiang, China)	AKT-1 antisense oligonucleotide	NCT05267899	Advanced solid tumors	
CDK-004	Exosomes	Codiak BioSciences (MA, USA)	Antisense oligonucleotide targeting STAT6	NCT05375604	Advanced hepatocellular carcinoma, gastric cancer metastatic to liver	
OTX-2002	Lipid nanoparticles	Omega Therapeutics (MA, USA)	Biscistronic mRNA downregulate c-Myc expression	NCT05497453	Hepatocellular carcinoma	

## 3. RNA Therapeutics and Gene-Based Delivery

RNA therapeutics are also emerging in cancer treatment. RNA therapeutics use RNA molecules, such as antisense oligonucleotides (ASOs), small interfering RNA (siRNAs), microRNAs (miRNAs), and aptamers. ASOs are short synthetic single-stranded nucleic acid sequences that can specifically bind to RNA target molecules in order to manipulate gene expression. ASO shows potential in cancer treatment by inhibiting the cell signaling pathways implicated in cancer progression, which potentially target oncogenes and growth factors in angiogenesis in cancer [47,48,49]. SiRNAs are short, double-stranded RNA molecules (typically 21–23 nucleotides) [50]. They are capable of stimulating mRNA degradation and reducing gene expression, specifically down-regulating expression of oncogenes [51]. miRNAs are endogenous, small non-coding RNAs that function in the regulation of gene expression [52]. The miRNAs regulate the oncogenes and tumor-suppressor genes, thus aiding in cancer therapy [53]. Short single-strand oligonucleotides known as aptamers have a high affinity and specificity for fitting targets and can develop secondary and tertiary structures. They are referred to as “chemical antibodies” and have the ability to target particular biomarkers for both therapeutic and diagnostic purposes in cancer [54].

Both viral and non-viral vectors can, in general, facilitate RNA delivery. Although designing adeno-associated viruses to transport nucleic acid cargo has generated a lot of interest in viral RNA delivery [55]. Nanoparticles are probably the most researched non-viral RNA delivery system. RNA encapsulation in nanoparticles physically shields nucleic acids from deterioration and, depending on the specific chemistry, may facilitate endosomal escape and cellular uptake [56]. Polymers are frequently utilized materials for distribution via nanoparticles due to their high level of chemical flexibility. The negatively charged RNA is often electrostatically condensed into nanoparticles using cationic polymers [57]. Naturally occurring polymers like chitosan and synthetic polymers like poly-L-lysine, polyamidoamine, and polyethyleneimine have all been used for RNA transport, with differing degrees of success [58]. Furthermore, some researchers have developed polymers especially for the delivery of nucleic acids. Particularly, poly(β-amino esters) have become widely used in DNA delivery because of their biodegradability and ease of synthesis, but they have also been shown to be able to transfer mRNA and siRNA [59]. Lipid-based nanocarriers have drawn a lot of interest for delivering RNA because of their demonstrated clinical success, high biocompatibility, and biodegradability. Lipid nanoparticles (LNPs), nanoemulsions, and liposomes are some of the most frequently studied lipid-based systems. These carriers are appealing platforms for systemic RNA distribution because they provide a safe environment for RNA molecules, extend circulation time, and increase cellular uptake [60].

## 4. Applications of Nanomedicine in Cancer Therapy

In this section, we discuss the utilization of nanomedicine in the treatment of cancer, specifically highlighting recent advancements in its application in chemotherapy, gene therapy, and immunotherapy.

### 4.1. Biological Nanocarriers in Chemotherapy

Nanoparticle-based chemotherapeutic drugs are mostly used to treat cancer due to their ability to accurately deliver pharmaceuticals in response to internal or external stimuli like pH, redox, radiation, and magnetic fields. Chemotherapy medications are delivered via chemical, inorganic, composite, and biological nanoparticles [61,62]. Among them, biological nanoparticles are important tools for drug delivery due to their safety, biocompatibility, rapid degradation, and targeting ability [63].

DNA-based NPs (Figure 4) are very advantageous because of their superior sequence programmability, biocompatibility, and degradability [64]. With the help of particular functional components, DNA-based NPs can be efficiently loaded with a range of medications, deliver pharmaceuticals to tumor tissues with precision, enhance cellular drug uptake, and provide stimulus-responsive drug release [65]. Interestingly, triangular DNA origami nanostructures, which are DNA-based NPs with particular hydrodynamic dimensions and geometries, can passively aggregate in tumor tissues and exhibit superior targeting [66], evident from Table 3. Functional components like aptamers enable DNA-based NPs to actively target chemotherapeutic medications to increase their efficacy and decrease side effects. The physiological characteristics of tumor tissues differ greatly from those of normal tissues. The designed DNA-based NPs cause structural reorganization, exposing the coated drug to the influence of various physiological characteristics or stimuli [67]. This improves the precise delivery and release of the drug, thereby enhancing its anti-tumor effect and mitigating damage to normal tissues.

The most common proteins used in nanocarrier designs are ferritin, albumin, transferrin, low-density lipoprotein, high-density lipoprotein, and so forth (Figure 4) [74]. Albumin-bound paclitaxel is the most well-known protein-based nanoparticulate delivery system (NDS) available [75]. For therapeutic use, the hydrophobic anticancer medication paclitaxel must be dissolved in polyoxyethylene castor oil, which might result in severe allergic responses in certain people. By means of hydrophobic contact, paclitaxel was incorporated into albumin to form albumin-bound paclitaxel nanoparticles, measuring 130 nm in diameter [76,77]. These nanoparticles are sold under the brand name Abraxane. The most prevalent serum protein in the human body, albumin, simply addresses allergies and enhances efficacy while lowering toxicity to healthy tissues and organs. It also lacks the immunological reaction triggered by castor oil. Furthermore, as research on protein-based NPs continues, stimuli-responsive NDSs have been developed to increase targeting efficiency. These NDSs can precisely deliver the medication to the tumor location and respond to stimuli in the tumor microenvironment, allowing for targeted therapy [76,77]. Iron homeostasis-related proteins, including ferritin and transferrin, are another family of proteins that are frequently built as NDSs. These proteins function very well in receptor-mediated active targeted delivery [78].

Currently, human ferritin (HFt) and other ferritin sources, such as iron-storing proteins, have been utilized as biological nanocarriers. Through TfR1-mediated specific targeting, heavy chain ferritin (HFn) can bind to tumor cells preferentially and internalize them quickly both in vitro and in vivo. Thus, therapeutic medications can be delivered into tumors selectively by HFn [79]. Ferritin’s intrinsic tumor-targeting capabilities and distinct nanocage structure make it an extremely desirable biological nanocarrier [79].

Lipoproteins are the naturally occurring spherical biomolecules composed of lipids and proteins. They serve as endogenous carriers for lipophilic substances and are often repurposed for drug delivery, particularly low- and high-density lipoproteins (LDL, HDL) due to their tumor-targeting properties [80]. Low-density and high-density lipoprotein (LDL and HDL) molecules are frequently utilized to transport chemotherapy drugs [80]. On the surface of tumor cells, LDL receptors are extensively expressed. These receptors supply the lipid matrix required for the creation of membrane systems in rapidly proliferating tumor cells. LDL can encapsulate a variety of hydrophobic loads, including radiolabels, photosensitizers, and different chemotherapeutic medications, as a delivery vehicle [81]. There are an increasing number of studies on recombinant/synthetic LDL, as natural LDL is obtained via plasma separation, which is challenging to create on a large scale. For instance, a pH-sensitive ApoB 100/oleic acid-doxorubicin/nanostructured lipid carrier NPs that exhibited strong suppression of breast cancer and enhanced accumulation at the tumor site [82].

HDL is a naturally occurring nanocarrier that has a structure like LDL. Contrary to LDL, which identifies receptors and facilitates the process of endocytosis, HDL specifically identifies scavenger receptor class B type 1 [83]. The primary protein of HDL remains on the surface of the cell membrane, while its core lipophilic cholesteryl ester directly enters the cell cytoplasm [80]. In addition, the primary apolipoproteins found in HDL have a smaller number of amino acids compared to those found in LDL. This characteristic helps prevent the creation of massive, irreversible aggregates [84]. All the aforementioned factors establish a solid foundation and advantageous circumstances for HDL as a means of transportation. The utilization of native HDL as a delivery mechanism is infrequent in current studies, but recombinant HDL (rHDL) nanoparticles can be employed for the transportation of chemotherapy medicines. As an illustration, Wang et al. developed rHDL NPs that contained doxorubicin and had a strong attraction to SR-B1. These nanoparticles were designed to be used in the treatment of liver cancer. Furthermore, rHDL can serve as a co-delivery vehicle for the combined administration of various chemotherapeutic medicines or the combined administration of chemotherapy and immunotherapy [85]. Rui et al. [86] created rHDL NPs for paclitaxel and doxorubicin, which significantly improved the accumulation of medicines in cancer cells and enhanced the antitumor response when used together at synergistic drug ratios. When compared to chemotherapy alone, it can be utilized to significantly enhance survival outcomes.

### 4.2. Biological Nanocarriers in Gene Therapy

Recently, advancements in gene manipulation technologies, such as gene silencing and gene editing, have allowed scientists to target specific genes and regulate their activity to treat various diseases. The general mechanism of nanoparticle targeting and cellular/sub-cellular drug delivery using nanomedicine (including gene therapy) is summarized in Figure 5. This approach has shown promising results and gained significant attention, particularly in the field of cancer therapy [87,88].

Tumor suppressor genes exert their inhibitory effect on cell growth when they are activated or overexpressed. Upregulation of one or more tumor suppressor genes can effectively impede the growth and advancement of malignancies. Both mRNA and plasmid can facilitate protein expression, but mRNA achieves this more rapidly and without the risk of mutation, integration, or other undesirable occurrences, making it a safer and more efficient method [89]. Nevertheless, RNA molecules exhibit a lack of stability and are not able to pass through membranes easily, necessitating the need for the development of delivery mechanisms. Kong et al. [12] developed a redox-responsive NP system that effectively transports mRNA expressing p53. This system successfully inhibited the growth of hepatoma and non-small cell lung cancer cells by triggering cell cycle arrest and apoptosis. mRNA carries a negative charge, so to effectively deliver mRNA drugs, most currently used nanoparticles (NPs) employ a positively charged carrier/inherent charge. These positively charged NPs can form stable complexes with mRNA, allowing for high loading rates. Examples of such NPs include ionizable lipid NPs, polymer–lipid hybrid NPs, and biocompatible biological nanostructures, among others [90,91,92].

Plasmid DNA-based gene therapy for cancer often targets apoptotic pathways to selectively induce cell death in tumor cells. By delivering plasmids encoding pro-apoptotic genes, which can trigger programmed cell death in cancer cells while minimizing harm to healthy tissues. For the delivery of plasmid DNA in gene therapy, non-viral transfection vectors provide a safer substitute for viral methods. These vectors, which include inorganic and polymeric nanoparticles, shield the plasmid DNA from deterioration and promote endocytosis, which is the process by which the DNA is taken up by cells. Once inside the cell, the DNA must escape the endosome and reach the nucleus to enable gene expression (Figure 6).

Gene suppression can be utilized as a treatment for cancer by selectively inhibiting the expression of specific genes that generate aberrant or detrimental proteins, employing techniques such as small interfering RNA (siRNA) therapy. Currently, there is considerable advancement in the field of nanocarriers for delivering siRNA. This includes the development of lipid nanocarriers, polymer nanoparticles, dendrimers, and inorganic nanoparticles [93,94]. Furthermore, the previously described biological nanoparticles can also transport siRNA with a substantial payload and excellent biocompatibility. Wang et al. [95] utilized DNA origami technology to create a DNA nanodevice capable of simultaneously delivering siRNA and doxorubicin (Nanodevice-siBcl2-si P-glycoprotein- doxorubicin). This nanodevice demonstrated strong cytotoxicity and effectively inhibited tumor development without causing any detectable systemic side effects. The CRISPR/Cas9 gene editing method, a form of cancer gene suppression therapy, has the ability to permanently eliminate tumor survival genes [96]. This overcomes the constraints of standard cancer therapy, such as the need for frequent dosing, and enhances the effectiveness of treatment [96].

### 4.3. Biological Nanocarriers in Immunotherapy

Immunotherapy has experienced significant advancements in recent years, reaching a state of maturity. Its introduction has brought about a revolutionary change in the standard approach to treating cancer. The primary focus of tumor immunotherapy is immune cells, which are responsible for triggering the body’s anti-tumor immune response. This response involves inhibiting negative immune regulators and enhancing the ability of immune cells to identify tumor cell surface antigens, ultimately leading to the elimination of tumor cells. An advancement of immune therapy is cellular backpacks, where the chemotactic sensitivity of immune cells with drug-loaded backpacks is employed for drug delivery [97,98,99]. During the initial phase of tumor immunotherapy, the primary approach is the direct targeting of tumor cells via cytokines generated by immune cells. As cancer immunotherapy research progresses, immune checkpoint inhibitors, tumor vaccination, and chimeric antigen receptor T cell therapy have emerged as the primary methods in immunotherapy [100].

Tumor cells have the ability to block the body’s immunological response, which allows them to avoid being eliminated by the immune system. They do this by producing chemicals that interact with T cells and prevent them from functioning properly. Nanomedicine is being utilized in several ways to enhance immune checkpoint inhibitors, hence augmenting their efficacy and transcending their limitations. NDSs have demonstrated their efficacy in decreasing the dosage of ICIs and managing immune-related adverse events [82]. According to Meir et al. [101], αPDL1-conjugated gold nanoparticles successfully inhibited tumor development at a dosage that was just 1/5th of the typical clinical dose. In addition, nanocarriers can be engineered as intelligent systems to regulate the release of drugs in response to specific stimuli found in the tumor microenvironment. This is anticipated to improve the effectiveness of nanoformulations in treating diseases.

DNA origami nanotechnology has enabled the creation of precise and multifunctional platforms for cancer treatment. One such device, DSWAC/siPD-L1, delivers doxorubicin, CpG, and siPD-L1 directly into tumor cells using a DNA origami scaffold functionalized with the AS1411 aptamer. This nanodevice protects siPD-L1 from degradation and ensures targeted release, while doxorubicin induces immunogenic cell death, promoting antigen presentation and dendritic cell activation. Simultaneously, siPD-L1 suppresses PD-L1 expression, reducing immune escape and enhancing T cell-mediated antitumor responses [102].

A similar nanodevice was engineered to co-deliver multiple immunostimulators, including doxorubicin, dsDNA, IL-12, and shPD-L1, using a DNA origami platform decorated with cRGD peptides for targeted delivery. Disulfide bonds enabled glutathione-sensitive cargo release within tumor cells. This strategy reprogrammed the tumor microenvironment, increased CD8+ and CD4+ T cell infiltration, and triggered strong antitumor immunity, significantly reducing tumor growth and lung metastasis without systemic toxicity [103].

Cancer vaccines consist of immunogenic elements, such as neoantigens and adjuvants, which are administered to antigen-presenting cells in peripheral lymphoid tissue. Enclosing immunogenic elements in nanocarriers can prevent the degradation of antigens and significantly enhance their stability [104]. Furthermore, nanovaccines have the ability to simultaneously encapsulate and distribute both antigens and adjuvants. This capability leads to a significant improvement in the immune response and effectiveness of vaccines, as demonstrated by Schijns et al. [105]. In addition, nano vaccines can effectively target immunological organs such as lymph nodes and spleen for delivery. Nano vaccines can transport a greater number of antigens from injection sites or tumors to lymph nodes, or from the blood to the spleen, by strategically manipulating physical properties (such as size, colloidal stability, electrostatic interactions, and deformability) or chemical properties (such as light, pH, and enzyme responsiveness). This is achieved through rational design [106]. Specifically, nano vaccines that have been modified with targeting ligands can be actively directed and transported to precise subregions of immune cells. An active lymphatic accumulation system utilizing click chemistry was developed to improve the targeted delivery of antigens and adjuvants to the lymphatic subcapsular sinus [107]. In the end, NPs have the ability to improve immune responses by providing sustained or regulated release capabilities.

## 5. Challenges and Regulatory Considerations of Cancer Nanomedicines

The intricate nature of nanomedicines, characterized by their diverse structures, makes it difficult to fully assess their physicochemical behavior, posing unforeseen translational challenges in some cases. This poses a challenge for regulatory review, particularly when comparing follow-on versions with the original reference product. The conventional model of using the sameness technique to ensure quality and bioequivalence in blood plasma is not suitable for nanomedicines, which require a different approach. The European Medicines Agency (EMA) and Food and Drug Administration (FDA) released reflection papers and draft guidance for the industry to address the issue of non-equivalence in authorized parenteral colloidal iron follow-on versions. These documents provide a platform for industry stakeholders to contribute their current thoughts on evaluating complicated products. It is logical to follow a systematic process to determine the level of similarity, starting with the evaluation of quality, including critical quality characteristics, and the assessment of nano properties. This should be followed by a non-clinical biodistribution assay, which is necessary in the European Union but not in the United States. Finally, clinical evaluation should be conducted. The collective body of evidence supporting the approval of subsequent iterations of nanomedicine is evaluated on an individual basis. The concept of interchangeability, often known as substitutability, poses a significant issue. Nevertheless, there is still a lack of a clearly defined or standardized approval process for these subsequent versions, which leads to potential discrepancies in the approval process. In order to advance, it is imperative to establish a science-focused forum where stakeholders and professionals in the field can engage in discussions.

## 6. Conclusions

The advancements in tumor therapy procedures present new prospects for the utilization of nanomedicine. Integrating intelligent nanodevices into tumor chemotherapy, gene therapy, and immunotherapy can enhance the efficiency of drug delivery, enabling targeted, precise, and controllable distribution. The translation of experimental anticancer nanomedicines into clinically feasible treatments continues to be a substantial challenge. Examples of applications include improving patient population stratification in clinical trials, optimizing the dosage of nanomedicine in combination therapy, and ensuring the production of high-quality and reproducible industrial nanomedicine. As nanotechnology research progresses, there is a prediction that the accuracy of scientific design and process engineering at the molecular level will surpass the basic technology used in NDSs research and development. This will open up new potential for NDSs.

## Figures and Tables

**Figure 1 biomolecules-15-00802-f001:**
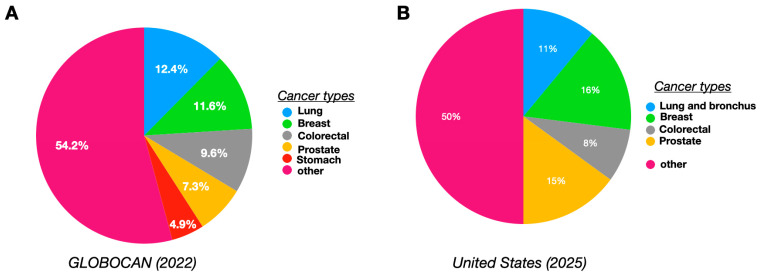
Cancer diagnosis trends (by cancer type) worldwide as per the Global Cancer Statistics: GLOBOCAN 2022 report (**A**) [1] and newly diagnosed cancer cases in the United States in 2025 (**B**) [2].

**Figure 2 biomolecules-15-00802-f002:**
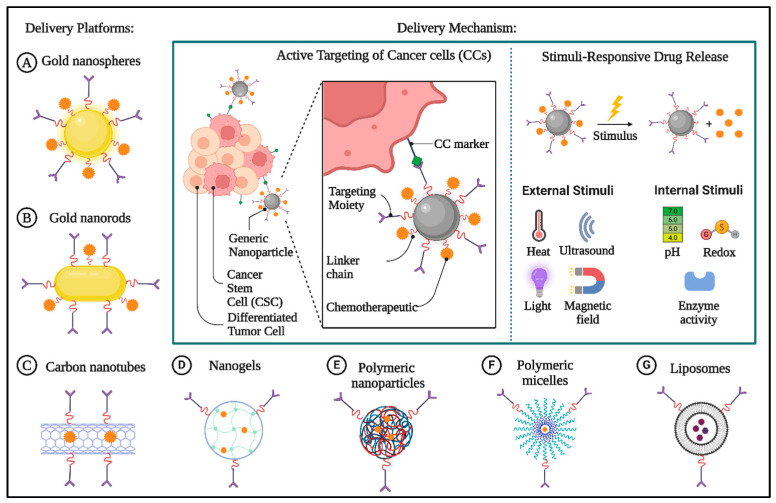
The diagram showcases various nanocarrier systems and their delivery methods for targeted cancer treatment. On the left side, it displays different nanocarrier systems employed for drug delivery. These systems are equipped with targeting moieties (depicted as purple Y-shaped structures) to improve their specificity for cancer cells. The middle section illustrates the active targeting mechanism for cancer cells (CCs), emphasizing how functionalized nanoparticles interact with cancer cell surface markers through targeting moieties. A chemotherapeutic drug is attached to the nanoparticle via a linker chain, allowing for accurate delivery to both cancer stem cells (CSCs) and differentiated tumor cells. The right section shows stimulus-responsive drug release, where external or internal triggers cause the therapeutic agents to be released from the nanocarriers. This figure highlights the strategic combination of intelligent nanomaterial design and controlled release systems to enhance therapeutic effectiveness, reduce off-target effects, and address tumor heterogeneity in cancer nanomedicine. (Created by using Biorender.com).

**Figure 3 biomolecules-15-00802-f003:**
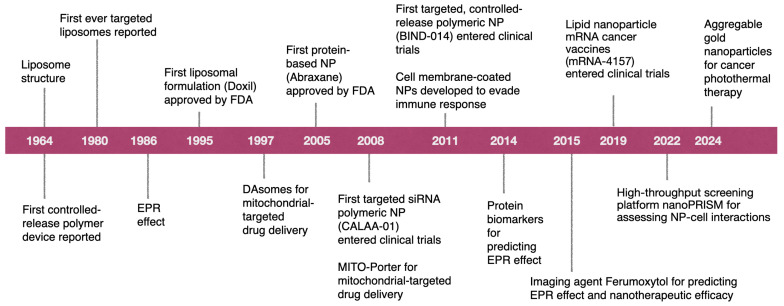
Graphical representation of the advances in cancer nanomedicine till 2024.

**Figure 4 biomolecules-15-00802-f004:**
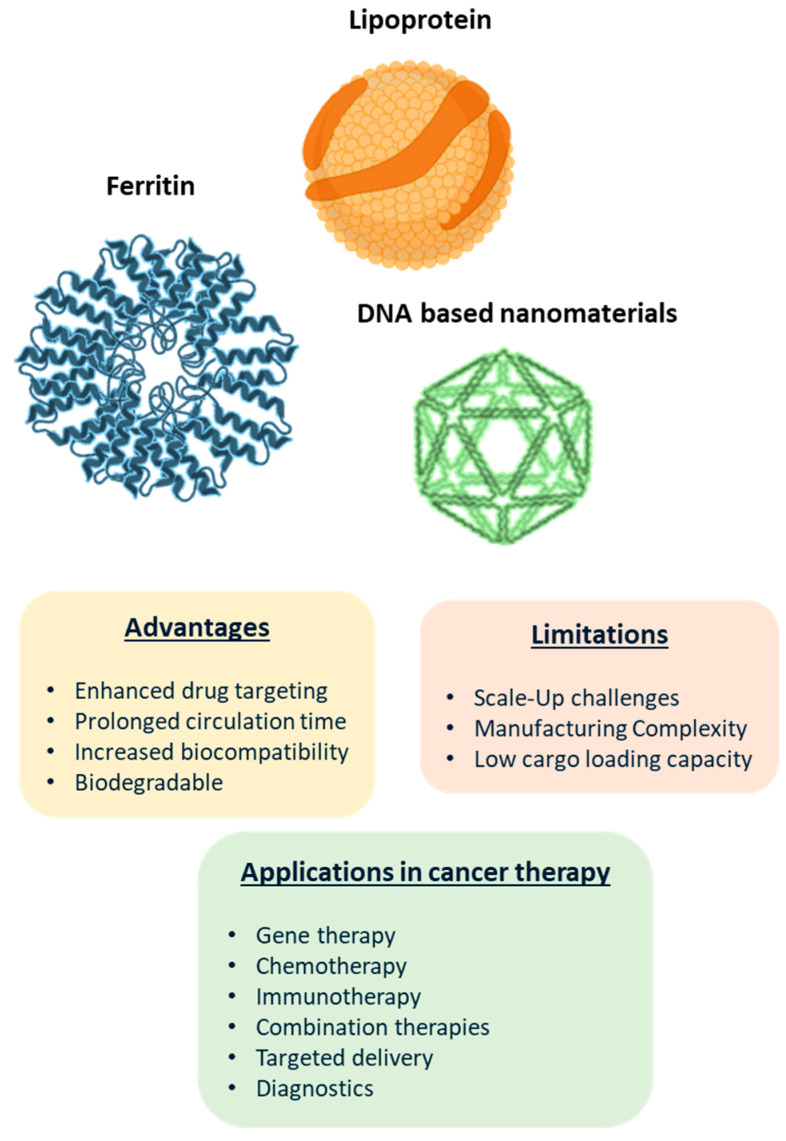
Biological and DNA-based nanomaterials in cancer therapy.

**Figure 5 biomolecules-15-00802-f005:**
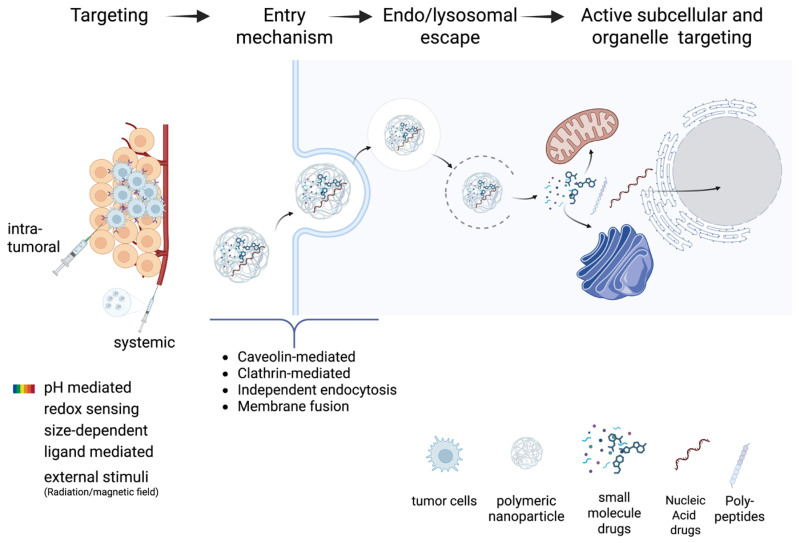
Mechanism of stimulus-responsive nanomedicine in gene therapy.

**Figure 6 biomolecules-15-00802-f006:**
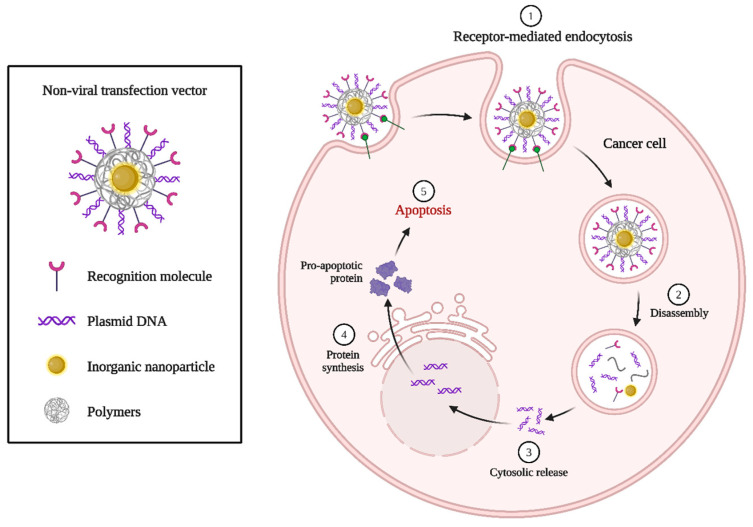
A schematic depiction of a non-viral gene delivery system utilizing a nanoparticle-based transfection vector for targeted cancer treatment through ligand-mediated targeting is presented. The inset on the left illustrates the makeup of the non-viral vector, which includes an inorganic nanoparticle (yellow core) enveloped by polymers (grey), plasmid DNA (purple), and surface recognition molecules (magenta) for targeting purposes. The right panel describes the intracellular trafficking and therapeutic action of the vector in five consecutive stages. (1) Receptor-mediated endocytosis begins the cellular uptake of the functionalized nanocarrier when it binds to specific receptors on the cancer cell surface. (2) Once internalized, the vector disassembles within the endosome, releasing the plasmid DNA and other components. (3) The DNA is then transported into the cytosol, where it escapes from endosomal confinement. (4) The plasmid DNA reaches the nucleus, where it initiates transcription and translation to produce therapeutic proteins. (5) The expression of pro-apoptotic proteins ultimately induces programmed cell death (apoptosis) in the cancer cell. This figure underscores the potential of non-viral vectors for safe, targeted, and effective intracellular gene delivery, offering a promising approach for cancer gene therapy while circumventing the immunogenicity and safety issues linked to viral vectors. (Created by using Biorender.com).

**Table 3 biomolecules-15-00802-t003:** DNA-based NPs and their effect.

DNA Nanostructures	Chemotherapeutic Drugs	Effect	Reference
DNA tetrahedron	Doxorubicin	Selectivity and inhibition of breast cancer cells and drug delivery	[68]
DNA icosahedron		Selective targeting	[69]
DNA octahedron		Efficient and specific internalization for killing epithelial cancer cells	[70]
DNA triangle and tube		Increased doxorubicin cellular internalization and elevated susceptibility to drug-resistant adenocarcinoma cells	[71]
RCA-based nanostructures		pH-Responsive Drug Delivery	[72]
DNA icosahedron	Platinum	Precise delivery of platinum nanodrugs to cisplatin-resistant cancer	[73]
DNA nanorod	Daunorubicin	Circumvent drug-resistance mechanisms in a leukemia mode	[64,71]

## Data Availability

Not applicable.

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
