# Peer review of "Biological Nanocarriers in Cancer Therapy: Cutting Edge Innovations in Precision Drug Delivery"

_biomolecules, 2025, doi:10.3390/biom15060802_

Round 1
Reviewer 1 Report
Comments and Suggestions for Authors
This review article provides a timely and relevant overview of the current landscape of biological nanocarriers in cancer therapy, focusing on cutting-edge innovations that promise enhanced precision in drug delivery. It effectively synthesizes a broad range of research, covering the applications of nanomedicine across various therapeutic modalities, including chemotherapy, gene therapy, and immunotherapy. The authors are to be commended for the generally clear and accessible writing style, which is further enhanced by the inclusion of well-chosen figures and tables that visually support the concepts presented. The article's comprehensiveness, incorporating recent references and presenting the information in a logically structured manner with clear headings, makes it a valuable resource for readers seeking an introduction to the field. Furthermore, the incorporation of illustrative examples, showcasing specific nanocarriers and their applications, greatly assists in solidifying understanding and demonstrating the potential of these technologies.
However, while the review successfully delivers a broad overview, it would significantly benefit from a more critical and nuanced analysis of the field. Currently, the manuscript leans toward a descriptive approach and could be strengthened by a more in-depth exploration of the inherent limitations, potential pitfalls, and persistent challenges associated with each of the discussed approaches. For instance, a more detailed discussion of off-target effects, long-term toxicity concerns, and the complexities of immune system interactions would add considerable value. Addressing these points will elevate the manuscript from a general overview to a more insightful and impactful contribution to the field.
A few specific questions to the authors:
- Update Figure 1, please use the same color coding in both graphs, it will help readers faster comparison and understanding.
- Please add DOIs for references 5 and 47.
Author Response
A few specific questions to the authors:
1. Update Figure 1, please use the same color coding in both graphs, it will help readers
faster comparison and understanding.
Response: The authors are thankful for the insights. Figure 1 has been revised as per the
suggestions
2. Please add DOIs for references 5 and 47.
Response: The authors are thankful for the comments. Reference #5 (PMID) and
reference #47 (now 77) (ISBN) have been incorporated
Reviewer 2 Report
Comments and Suggestions for Authors
The authors here aim to present a review on the use of "Biological nanocarriers in cancer therapy: Cutting edge innovations in precision drug delivery ". My comments are as follows:
- The figures are well created and provide information, however Figure 4 could be modified to make it look more professional and provide more relevant scientific information
- Authors should include a section on RNA therapeutics and gene based delivery
- Although the focus of their review is "biological nanocarriers" a number of lipid and polymer based NPs are being used in the clinics in conjugation with both passive and active targeting, the authors could add a few small paragraphs and a brief illustration on these classes as well as Carbon Dots.
- The following references should be added:
https://onlinelibrary.wiley.com/doi/abs/10.1002/anie.200502113
https://www.science.org/doi/full/10.1126/sciadv.aaz6579?rfr_dat=cr_pub++0pubmed&url_ver=Z39.88-2003&rfr_id=ori%3Arid%3Acrossref.org
https://pubmed.ncbi.nlm.nih.gov/31183964/
Author Response
The authors here aim to present a review on the use of "Biological nanocarriers in cancer therapy: Cutting edge innovations in precision drug delivery ". My comments are as follows:
- The figures are well created and provide information; however Figure 4 could be modified to make it look more professional and provide more relevant scientific information
Response: The authors are thankful for the insights. Figure 4 has been revised as per the suggestions
- Authors should include a section on RNA therapeutics and gene-based delivery
Response: The authors are highly thankful for the comments. A separate section has been added (section 3) to the manuscript and was highlighted in yellow.
- Although the focus of their review is "biological nanocarriers" a number of lipid and polymer-based NPs are being used in the clinics in conjugation with both passive and active targeting, the authors could add a few small paragraphs and a brief illustration on these classes as well as Carbon Dots.
Response: The authors are thankful for the valuable suggestion, a discussion on NPs and active and passive targeting has been incorporated into section 2 and highlighted in yellow (line: 141-192).
- The following references should be added:
https://onlinelibrary.wiley.com/doi/abs/10.1002/anie.200502113
https://www.science.org/doi/full/10.1126/sciadv.aaz6579?rfr_dat=cr_pub++0pubmed&url_ver=Z39.88-2003&rfr_id=ori%3Arid%3Acrossref.org
https://pubmed.ncbi.nlm.nih.gov/31183964/
Response: The authors are thankful for the comments, the suggested references (# 22, 29, 99) have been included in the manuscript.